# Systematic review of the characteristics of brief team interventions to clarify roles and improve functioning in healthcare teams

**Kelley Kilpatrick**[1,2]*, **Lysane Paquette**[3], **Mira Jabbour**[2], **Eric Tchouaket**[4], **Nicolas Fernandez**[5], **Grace Al Hakim**[6], **Véronique Landry**[3], **Nathalie Gauthier**[7], **Marie-Dominique Beaulieu**[8], **Carl-Ardy Dubois**[9]

**1** Susan E. French Chair in Nursing Research and Innovative Practice, Ingram School of Nursing, Faculty of Medicine, McGill University, Montréal, Québec, Canada, **2** Centre intégré universitaire de santé et de services sociaux de l'Est-de-l'Île-de-Montréal-Hôpital Maisonneuve-Rosemont (CIUSSS-EMTL-HMR), Montréal, Québec, Canada, **3** Faculty of Nursing, Université de Montréal, Montréal, Québec, Canada, **4** Department of Nursing, Université du Québec en Outaouais, Saint-Jérôme, Québec, Canada, **5** Department of Family Medicine and Emergency Medicine, Faculty of Medicine, Université de Montréal, Montréal, Québec, Canada, **6** Clinical and Professional Development Center, American University of Beirut Medical Center, Beirut, Lebanon, **7** Nursing and Physical Health Directorate, Centre intégré universitaire de santé et de services sociaux de la Capitale-Nationale, Québec, Québec, Canada, **8** Faculty of Medicine, Université de Montréal, Montréal, Québec, Canada, **9** Department of Management, Evaluation and Health Policy, School of Public Health, Université de Montréal, Montréal, Québec, Canada

* kelley.kilpatrick@mcgill.ca

**Data Availability Statement:** All relevant data are within the paper and its Supporting Information files.

## Abstract

### Aim

Describe brief (less than half a day) interventions aimed at improving healthcare team functioning.

### Methods

A systematic review on brief team interventions aimed at role clarification and team functioning (PROSPERO Registration Number: CRD42018088922). Experimental or quasi-experimental studies were included. Database searches included CINAHL, Medline, EMBASE, PUBMED, Cochrane, RCT Registry-1990 to April 2020 and grey literature. Articles were screened independently by teams of two reviewers. Risk of bias was assessed. Data from the retained articles were extracted by one reviewer and checked by a second reviewer independently. A narrative synthesis was undertaken.

### Results

Searches yielded 1928 unique records. Final sample contained twenty papers describing 19 studies, published between 2009 and 2020. Studies described brief training interventions conducted in acute care in-patient settings and included a total of 6338 participants. Participants' socio-demographic information was not routinely reported. Studies met between two to six of the eight risk of bias criteria. Interventions included simulations for technical skills, structured communications and speaking up for non-technical skills and debriefing.

**Funding:** KK, LP, MJ, ET, NF, VL, NG, MDB, CAD received the award from Canadian Institutes of Health Research Program: Project Grant Competition: 201603PJT Application Number: 363528 https://cihr-irsc.gc.ca/e/193.html The funders had no role in study design, data collection and analysis, decision to publish, or preparation of the manuscript.

**Competing interests:** The authors have declared that no competing interests exist.

Debriefing sessions generally lasted between five to 10 minutes. Debriefing sessions reflected key content areas but it was not always possible to determine the influence of the debriefing session on participants' learning because of the limited information reported.

## Discussion

Interest in short team interventions is recent. Single two-hour sessions appear to improve technical skills. Three to four 30- to 60-minute training sessions spread out over several weeks with structured facilitation and debriefing appear to improve non-technical skills. Monthly meetings appear to sustain change over time.

## Conclusion

Short team interventions show promise to improve team functioning. Effectiveness of interventions in primary care and the inclusion of patients and families needs to be examined. Primary care teams are structured differently than teams in acute care and they may have different priorities.

## Introduction

There is a growing interest worldwide to understand how to improve team functioning and team performance in healthcare settings [1]. Poor team functioning has been identified as a critical factor of adverse events of patient safety [2]. Globally, four out of 10 patients in primary and ambulatory care are harmed [3] 134 million adverse events occur in hospitals contributing to 2.6 million deaths; and medication errors cost an estimated 42 billion USD annually [4]. In Canada, it is estimated that preventable patient safety incidents occur every minute and 18 seconds [5]. Several national and international reports [6–9] highlight that improved team functioning lead to better outcomes for patients, providers and healthcare systems.

In their seminal review, Cohen and Bailey [10] defined a team as a group of two or more people, who are interdependent in their respective tasks and share common goals and responsibilities for results. Team functioning is influenced by processes that included decision-making, communication, cohesion, care coordination, problem-solving and focus on patients and families [11]. Mathieu et al. (2019) [12] updated their review of the team effectiveness literature in organizational research conducted in the last 10 years. They identified 29 meta-analyses including 30 structural and process factors that predicted team effectiveness [12]. They argued that team effectiveness is a multi-dimensional and complex construct where effective teams navigate between different structures, mediating mechanisms (e.g., processes), and external influences to efficiently produce tangible outputs that are high quality [12]. In healthcare, teams rely on the contribution of many professionals with different expertise to meet the increasingly complex needs of the population [13–15]. Team training is seen as essential to improve team performance [16, 17].

Role clarity between providers has been identified as an important factor to improve team functioning [16, 18, 19]. The lack of role clarity, lack of understanding of the boundaries between roles, and poorly defined scope of practice can jeopardize teamwork [20, 21]. Such problems are particularly salient given the context of healthcare reforms and system restructuring [19]. According to Hudson et al. [22], role understanding is an integral part of teamwork because it generates trust and mutual respect. Greater understanding of others' roles in

the team promotes role clarity to foster optimal utilization of all professional roles and improve patient outcomes and health system cost-effectiveness [23]. Hence, role clarity is key to effective team training interventions.

Teams are active learning systems where individuals develop relationships and apply knowledge to solve problems [24]. McEwan et al. (2017) [25] completed a systematic review and meta-analysis of teamwork training and interventions (n = 51). These authors identified four types of interventions for teamwork including didactic lectures/presentations, workshops, simulations, and on-site review activities. McEwan et al. (2017) [25] determined that teamwork interventions exerted a moderate effect on teamwork and team performance. However, approximately two-thirds of the teams identified by McEwan et al. (2017) [25] were outside of healthcare and included academia and experimental laboratory research.

Marlow et al. (2017) [26] completed a systematic review to examine team training interventions in healthcare (n = 197) and found that team training included a variety of training methods to address the needs of a wide range of care providers. The most frequent interventions identified in the review centered on improving team processes such as teamwork, awareness of the environment, leadership, shared understanding, decision-making, communication, coordination and team role knowledge. These researchers did not identify interventions lasting less than one day.

Team-based interventions where members are engaged are more effective [25]. In addition, interventions are more effective if they target several dimensions of teamwork simultaneously and are specific to the setting [25]. Sidani and Braden (2011) [27] defined interventions as rational actions and interrelated behaviours directed toward addressing a specific aspect of a problem to achieve a common goal [27]. These authors highlighted that interventions vary in their level of complexity from simple to complex. Complex interventions are made of several components and interrelated parts [27, 28]. When examining interventions, researchers [1, 27] have noted key characteristics to consider included the dose (e.g., duration, frequency), mode of delivery (e.g., written, verbal), and type of intervention.

As indicated above, there is consensus in the literature on the dynamic nature of healthcare teams, their contributions to quality of care and how longer team interventions can improve team functioning. However, as clinical loads continue to increase, due to greater complexity of health problems, ageing population and severe limitations imposed on resources, longer team training is less and less attractive. There is thus a growing need to envision short term interventions that can provide needed support and have an impact on team performance. Our team aims to address this gap in our understanding and describe the characteristics of brief (less than half a day) team interventions that contribute to improving team functioning.

## Materials and methods

We conducted a systematic review to describe the characteristics of brief team interventions to clarify roles and improve functioning in healthcare teams.

### Search strategy

The research targeted experimental or quasi-experimental studies published or pre-published between January 1990 and April 2020. The databases explored included CINAHL, Medline, EMBASE, PUBMED, Cochrane, RCT Registry-1990 to April 2020. Records were retrieved on April 21$^{st}$ 2020. A search for existing systematic reviews in the Cochrane Database and Prospero Registry was conducted. The gray literature was explored using the strategies proposed in Grey Matters (2014) [29], notably via the ProQuest, GraySource Index and Google Scholar databases. Searches were also conducted to find abstracts or conference proceedings and pre-

publications. In addition, the reference lists of selected papers were examined to identify additional studies. We worked with an academic librarian to develop and validate the search strategy and identify keywords for each database. Search strategies are provided in the Appendix. No language restriction was applied.

## Inclusion and exclusion criteria

We included randomized controlled trials (RCTs), experimental and quasi-experimental designs because we were looking to identify the characteristics of team interventions that were known to be effective. We retained systematic reviews (with or without meta-analysis) to conduct a hand search of the reference lists. An expanded search to include other research designs (e.g., observational study) was not necessary given the number of studies that were identified.

We included all studies where the intervention lasted less than a half day or 4.5 hours using experimental and quasi-experimental designs. We included teams in different contexts, within and outside of healthcare. Interventions developed for healthcare teams could be in primary and acute care, and include providers such as physicians, medical specialists, nurses, nurse practitioners, nurse clinicians, nursing assistants, licensed practical nurses, social workers, physiotherapists, occupational therapists, pharmacists, support personnel (e.g., secretaries, clerks), and patients and families. Primary care was defined as comprehensive healthcare services for common health concerns at the point of entry to the healthcare system [30]. Acute care included in-hospital or specialized ambulatory care [31].

We excluded studies where the intervention lasted more than a half day or 4.5 hours. The primary aim of the review was to identify effective short team interventions. As proposed by Higgins et al. (2019), we excluded observational and longitudinal studies and as well as qualitative methodologies as these studies are at increased risk of bias [32].

## Intervention

We retained interventions that influenced team functioning or team processes. Interventions could be geared to different members of the healthcare team, patients, families, managers or support staff. Data were extracted to determine key characteristics of the interventions including setting, duration, type of intervention, frequency and sequence of activities. Comparators and control conditions included no intervention or the usual functioning of the team.

## Study selection

The Cochrane handbook for systematic reviews of interventions served as a guide for this systematic review [33]. A review protocol was developed and published with PROSPERO (Registration Number: CRD42018088922) [34]. Training sessions were conducted with all assessors (n = 8) to review inclusion and exclusion criteria, the screening instrument and answer questions. All the publications identified following the application of the search strategy were uploaded into the Endnote reference management software and duplicates were removed. Subsequently, titles and abstracts, if available, were reviewed independently by two reviewers using the RAYYAN web application to exclude articles that were not relevant considering the inclusion criteria [35]. Full texts were reviewed if abstracts were not available.

Full-text review was undertaken for articles that met the inclusion criteria. Reviewers independently assessed if they met the inclusion criteria and a final decision was made about their inclusion in the systematic review. A third researcher (KK) acted as a tie-breaker in case of disagreement between reviewers. A PRISMA flow chart was generated to demonstrate the steps for selecting studies and document the reasons for exclusion [36]. Assessor agreement on all

inclusions and exclusions was 90.5%. Using Cohen's kappa, we obtained substantial inter-rater agreement at 61% across the eight assessors paired two by two [37].

### Data extraction

The data extraction form was developed using the recommendations of Kennedy et al. (2019) [38], and pilot-tested with extractors. Data from the retained articles were extracted by one reviewer and checked by a second reviewer independently. The extractions were integrated into a table to identify first author and year, country, characteristics of participants, characteristics of the intervention (e.g., duration, type), data collection instruments, use of a theoretical/conceptual framework, design, risk of bias, results, limits, strengths and funding sources. When more than one paper was published for the same study, data were extracted using one form and color coded to link the extraction back to the relevant article.

### Risk of bias assessment

We used the risk of bias assessment tool proposed by Kennedy et al. (2019) [38]. This tool was selected because it allowed us to assess study rigour in randomized and non-randomized intervention studies. The instrument includes eight items (i.e., cohort, pre/post comparison group, pre/post intervention data, random assignment of participants to intervention, random selection of participants for assessment, follow-up rate of 80% or more, comparison groups equivalent on socio-demographics, comparison groups equivalent at baseline on outcome measures). If a criterion was met, a score of one was indicated. If the criterion was not met, a score of zero was indicated. If the information provided did not allow the reviewer to assess fulfillment of the criterion, NR was indicated for not reported. If a criterion was not applicable because of the study design, we indicated NA. As proposed by Kennedy et al. (2019) [38], the NAs and NRs were assigned a zero to indicate that the criterion was not met. The instrument's inter-rater reliability using Cohen's kappa was moderate to substantial (0.41 to 0.80) for all items [38]. To gain a better understanding of the strength and gaps of the knowledge base, a total score was calculated by item and overall for each study. The highest possible score was eight. If no psychometric properties were reported for the instruments used in the studies, we searched the literature to determine the psychometric properties of the instruments. These papers are listed in the table.

### Analysis

A narrative synthesis was undertaken. No meta analysis or subgroup analysis was conducted because of the diverse characteristics of the interventions and practice settings.

## Results

The searches yielded 1712 unique records of which 1505 were excluded during title and abstract review. Following full text review of the remaining 207 papers, 187 were excluded based on reasons listed in Fig 1. Ultimately, the search yielded 20 papers [39–53] reporting on 19 studies. One study was reported in two papers [49, 50]. Fernandez et al. (2020) [54] reported on the development of the intervention used in their study (Rosenman et al., 2019). All the manuscripts were published in English. The retained studies were published between 2009 and 2020 (see Table 1). Studies were conducted in Australia [39], Belgium [55], France [56], Germany [40, 57], New Zealand [41], United States [42–52, 54], Singapore [58], and Taiwan [53]. Key study characteristics are presented in Table 1 and outlined below.

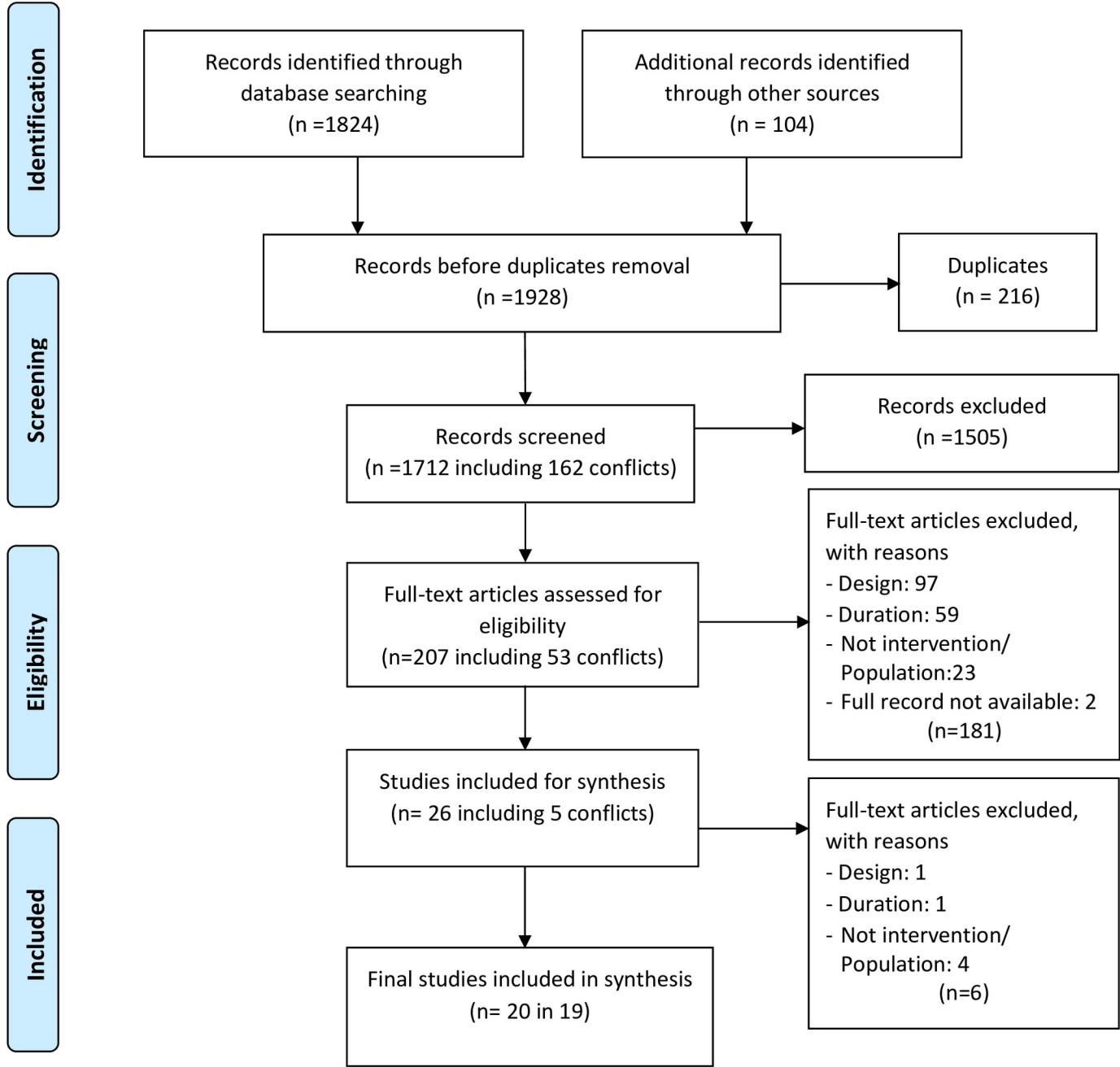

**Fig 1. PRISMA 2009 flow diagram\*.**

\* Moher D, Liberati A, Tetzlaff J, Altman DG, The PRISMA Group (2009). Preferred Reporting Items for Systematic Reviews and Meta-Analyses: The PRISMA Statement. PLoS Med 6(7): e1000097. doi:10.1371/journal.pmed1000097

**Table 1. Overview of included studies.**

| First Author, Year, Data Collection, Country | Study Design | Population Characteristics | Instruments Y if validated | Frame-work | Intervention | Results |
|---|---|---|---|---|---|---|
| Barzallo Salazar, 2014 [42] 2011 to 2012 United States | RCT | **Setting:** OR **Population:** Medical students beginning their obstetric rotation **Gender** (male; female) I: 12/28; 16/28 C: 12/27; 15/27 **Age** (years) Mean: I: 26; C: 25 | Personality tests General Decision Making Scale: Y Self-Construal Scale: Y | N | Simulated surgeries with training in basic surgical techniques and speaking up with trainees who witnessed a surgical error. The simulation was spread over two days and not more than one week between the surgical training session. A senior surgeon took a few minutes to create an environment for trainees to encourage/ discourage speaking-up using a scripted scenario | The trainees in the encouraged group were more likely to speak-up about the surgical error (p <0.001). The surgeon's attitude influenced trainees' willingness to speak up after controlling for personality traits (p <0.001). |
| Beck, 2019 [57] February to December 2017 Germany | RCT | **Setting:** simulated in-hospital cardiac arrest during mandatory BLS training at a University Medical Center **Population:** Physicians, nurses, scientist, administrative staff **Gender** Female % (n): I: 68 (160/235) C: 67 (116/174) **Age** (years) % (n) 16–2: 24 (57/241) C: 17 (30/178) 30–39: I: 34 (83/241) C: 33 (59/178) 40–49: I: 18 (44/241) C: 21 (38/178) 50–59: I: 20 (49/241) C: 24 (42/178) 60+: I: 3 (8/241) C: 5 (9/178) | German version of Team Assessment Scale (TAS): Y Internal consistency Cronbach's α 0.67–0.81 [59] | Y Salas framework and shared mental models | Intervention included a 90-minute training session with a lecture on outcome relevant actions, group work to establish priorities in case of arrest, 4 min video of in-hospital BLS and practical training in AED use in teams with feedback. Participants received hands-on training in a two-rescuer BLS scenario on a high-fidelity manikin and feedback using learning conversation and a performance checklist. | TAS was high in all dimensions. Overall score for BLS performance was not significantly different between the groups p = 0.49. No significant difference between groups: Team Adjustment Behavior (TAB) p = 0.82 |
| | | | | | | Cooperation and information exchange (CIE) p = 0.43 Team coordination (TC) p = 0.88 Hands-off time lower in the intervention group (5.4% vs. 8.9, p = 0.029). All dimensions of the TAS correlated negatively with the hands-off time (TC: CC = 0.23; p = 0.010, CIE: CC = 0.28, p = 0.001, TAB: CC = 0.28; p = 0.001). |
| Chang, 2019 [53] October 2015 to January 2016 Taiwan | Pre/post test | **Setting:** Large teaching hospital. **Population:** newly registered postgraduate trainees (residents, nurses, respiratory therapists) | **The technical skills:** Y **Non-technical skills (ANTS):** Y Cronbach alpha (α) values from 0.79–0.83 [81]. | N | Simulated transport of patient in sceptic shock with equipment difficulties and physiologic instabilities. The intervention included a two-hour training session spread over three months and included monthly in-situ scenarios, video-based feedback and focus group discussions to enhance re-evaluation, communication, prioritization of interventions, and equipment recovery. Tasks and competencies were outlined for each professional group. | Teams exhibited higher levels of non-technical skills (i.e., task management, teamwork, situational awareness, and decision-making) before and after transport (p values between 0.006 to 0.032), and participation in problem-solving (p values between 0.005 to 0.011). Only the results for the respiratory therapist group were not significant for participation in problem solving (p = 0.06). No corrections were applied for multiple comparisons in this study. |
| | | **Gender:** Not reported | | | | |
| | | **Age:** Not reported | | | | |
| Coppens et al., 2018 [55] February to April 2015 Belgium | RCT | **Setting:** High-fidelity simulation training in a Simulation Training Centre. **Population:** Nursing undergraduates N = 116 in 30 groups (3-5/ group): I: 15 groups (n = 60) C: 15 groups (n = 56) **Gender** Women %(n): I: 82 (49) C: 73 (41) Men: I: 18 (11) C: 27 (15) **Age** (years) % (n) I: 20–21 55 (33) >21: 45 (27) C: 20–21: 57 (32) >21: 43 (24) | Teamwork (CTS): (Kappa .78; interclass correlation .98) [82]: Y Team efficacy (TE): Construct validity: (Cronbach's a > .8) and internal consistency (r = .57, p < .0001) [83]: Y General Self-Efficacy Scale (SE): Cronbach's Alpha: .76–.90; [84]: Y Technical skills (TECH): N | Y | The 90-minute intervention included a 30-minute course on crisis resource management (CRM) principles with 45 minutes facilitated debriefing. Simulation mirrored a patient's room. Two scenarios lasting 15 minutes were completed. Debriefing using Steinwachs' approach included examining impressions following simulation, reconstructing the scenario, reflections on successes, challenges and ways to improve. | The intervention group had significantly higher scores on Teamwork (p = .011), CTS (p = .011), TE (p < .001) and TECH (p = .014), and a significant increase in all variables (SE (p = .02), CTS (p < .001), TE (p < .001)) except for TECH (p = .607). The experience from both interventions led to a significant increase in only CTS (p < .001) and TE (p = .001) for the control group. |

*(Continued)*

**Table 1.** (Continued)

| First Author, Year, Data Collection, Country | Study Design | Population Characteristics | Instruments Y if validated | Frame-work | Intervention | Results |
|---|---|---|---|---|---|---|
| Evain et al. 2019** [56] November 2015 to June 2016 France **Unclear if outcomes included 17 or 21 teams in the intervention group. | RCT | **Setting:** Scenarios in the emergency room; operating theatre; delivery suite; intensive care unit; and intra-hospital patient transport. **Population:** Year 1–5 trainees in anaesthetia and intensive care. **Gender:** n (%) In pairs FF/FM/MM I: 4(19)/ 11(52)/6 (29) C: 6 (29)/ 11 (52)/4 (19) **Age** Median [range] I: 27 [24–37] C:27 [24–30] | 1) Twelve scenario checklists: N 2) Ottawa global rating scale [85]: Y 3) Visual analogue scale (VAS): N 4) Cognitive appraisal ratio: N | N | The intervention included a 4-minute period for a team planning discussion. The discussion initiated by two standardized questions, namely 'Given the information provided, what can be expected?' and 'How will you organise yourselves?'. Facilitator prohibited from answering questions or leading the discussion. Oral briefing given before initiating the scenario. Following each simulation, two instructors led a structured debriefing. No details provided. | Clinical performance scores were higher in the intervention group (p = 0.039). After controlling for the scenario, the intervention associated with a 5-point (11%) increase in clinical performance score (95%CI 0.6–9.6), p = 0.029. No significant difference noted in crisis resource management scores following planning discussion (p = 0.065). Authors report similar perceived stress levels between the groups at three measurement times. |
| Fernandez, 2013 [52] August 2010 to March 2010 United States | RCT | **Setting:** Patient crisis resuscitation scenario **Population:** Code team. 4th-year medical students and 1st-, 2nd-, 3rd year residents in emergency medicine **Gender n (%):** Male I: 74 (63) C: 67 (60) **Age:** Mean(SD) I: 27.7 (3.16) C: 27.2 (2.94) | Checklists measures using evidence-based guidelines. N | Y | Two-hour computer-based teamwork training that included audio-narrated slide presentation viewed at individual workstations, video-recorded validated high-fidelity simulations for resuscitation scenarios of a cardiac arrest or hemorrhagic shock, and debriefing. | The intervention significantly increased teamwork behaviours and patient care behaviours in teams receiving the computer-based teamwork training intervention (F (1, 42) = 4.66, p reported as less than 0.05) after controlling for experience using a low-intensity simulation platform. Team size did not significantly affect teamwork or patient care behaviours. No details provided for the debriefing. |
| Fernandez, 2020* [54] April 2016 to December 2017 United States *Some inconsistencies noted between the abstract and main text. Data extracted from main text. | Single-blind RCT | **Setting:** Actual trauma resuscitation at a regional, university-affiliated level 1 trauma center. **Population:** 2nd- and 3rd- year emergency medicine and general surgery residents acting as trauma team leaders as part of their training. **Gender** n (%): Male C: 21/30 (70) I: 19/30 (63) **Age (years):** Mean (SD) C: 29 (2) I: 30 (3) **Residency year:** n (%) Postgraduate Y2 C: 14/30 (37) I: 19/30 (63) Postgraduate Y3 C: 16/30 (53) I: 11/30 (37) **Specialty:** n (%) Emergency medicine C: 19/30 (63) I: 26/30 (87) General surgery C: 11/30 (37) I: 4/30 (13) | **Team leadership measure:** N **Patient care measure** checklist: N. **Injury Severity Score** [ISS] [26]. Y | N | Intervention included a single, 4-hour session with facilitated discussion of trauma leadership skills (30–45 min), a didactic session on leadership behaviors in trauma care (30 min. lecture), simulations, and debriefing sessions. Simulations could be adapted to facilitate learning and meet core training requirements. During the simulation, one participant functioned as the team leader, while the second participant observed using a leadership checklist. Debriefing immediately followed each simulation. Three self-identified areas for improvement and instructor observations informed subsequent simulations. A plan was created for each participant to apply learning in practice. | Simulation-based leadership intervention resulted in a 56% improvement in leadership behavior after controlling for subject and patient factors (p<0.001). Intervention improved 5 out of the 7 leadership behaviors: Explicitly assuming leadership (p = 0.002); Performing pre-briefs (p < 0.001); Performing an arrival brief (p = 0.004); Performing huddles (p = 0.001); Seeking input (p = 0.030); Planning (p = 0.257); and Role assignment (p = 0.084). No significant differences in patient care between groups (p = 0.99)**. Leadership behaviors predicted patient care (p < 0.001) after controlling for experimental condition, year in residency, days since/until training, and ISS. Leadership behaviors appear to mediate the effect of training on patient care with a significant indirect effect. |

*(Continued)*

**Table 1.** (Continued)

| First Author, Year, Data Collection, Country | Study Design | Population Characteristics | Instruments Y if validated | Frame-work | Intervention | Results |
|---|---|---|---|---|---|---|
| Jankouskas, 2011 [43] Dates Not reported 10-month study period United States | Pre/post test | **Setting:** Intensive care unit transport team **Population**: 4-member teams of senior-year nursing students + third-year medical student**. Gender:** Male6% of the nursing student sample; 52% of the medical student sample **Age**: Not reported | University of the West of England Inter-professional Question naire Anesthetists' Non-Technical Skills (**ANTS**) : Y | Y | Three-hour training session with two video-recorded scenarios, a high-fidelity simulator and didactic material for crew resource management (CRM) training related to task management, teamwork, situational awareness, a review of basic life support, and facilitated debriefing of patient crisis management using a non blaming technique. Four-member teams included senior-year nursing students and third year medical students. | Significant differences were noted for task management (p = 0.05), teamworking (p = 0.02), and situation awareness (p = 0.01). No differences were noted in error rates, response time for oxygen placement, response time for bag-mask-valve ventilation (BMV), and response time for chest compressions. Correlations between CRM training and team effectiveness measured using error rate and response time were not significant. Team process and team effectiveness improved in all groups from pretest to posttest as an effect of team practice (p < .001). No details provided of participant views of facilitated debriefing. |
| Kalisch, 2015 [44] Dates Not reported United States | Pre/post test | **Setting:** medical–surgical patient virtual patient care unit in an academic health center **Population**: Nursing staff who provide direct care to patients (RNs and nursing assistants). **Gender:** 81% female (n = 35) **Age**: Not reported | **Nursing teamwork survey (NTS):** Y **Teamwork Knowledge Survey**: Y Concurrent, convergent, and contrast validity is strong [60].<br><br>Adapted TeamSTEPPS Questionnaire [86]: Y Computer and virtual experience: questionnaire: N | Y | One hour and 40 minute intervention included a 30-minute podcast of teamwork followed by a one-hour virtual simulation using a multi-user virtual environment. A 10-minute debriefing with an experienced trainer was conducted to highlight teamwork behaviours, provide feedback on the scenarios, and examine what would be done differently in the future. Three scenarios highlighted ways to resolve team conflicts between nurses and nursing assistants for common nursing problems using eight teamwork behaviours consistent with the Salas TeamSTEPPS model. The modules for the virtual environment were purchased from a software developer. The intervention required extensive preparation to develop the virtual unit, conference room, and semi-private rooms for patients. | Scores for teamwork overall (p = 0.12), trust (p = 0.042), team orientation (p = .004), and backup (p = .045) improved significantly. Scores for shared mental model, team leadership and teamwork knowledge did not reach significance. Computer proficiency pre- and post-intervention did not influence scores. No details provided of the results of the debriefing exercise. |

*(Continued)*

**Table 1.** (*Continued*)

| First Author, Year, Data Collection, Country | Study Design | Population Characteristics | Instruments Y if validated | Frame-work | Intervention | Results |
|---|---|---|---|---|---|---|
| Liaw, 2019 [58] Data collection dates not reported. Singapore | Three-arm RCT | **Setting:** Healthcare course involving three universities. Interprofessional bedside rounds involving a simulated patient with physical and psychosocial issues at a university simulation center. Students logged in to a virtual platform. **Population:** Healthcare students from medicine, nursing, pharmacy, physiotherapy, occupational therapy, and social work as part of their course work. **Gender:** Female: 65% **Age:** Not reported | Team performance rating scale: N Attitudes Towards Interprofessional Health Care Teams (ATIHCT). Cronbach α: 0.82. Y. Interprofessional Socialization and Valuing Scale (ISVS). Cronbach α: 0.95. Y. | N | Intervention 1 lasted 30 minutes and included asynchronous delivery of didactic training on cognitive tools to use in interprofessional rounds. Online video covered a modified ISBAR communication tool on team member roles, sequence and nature of communication with patients, families and healthcare team members, and the biopsychosocial model of health to facilitate the development of an actionable plan of care. Intervention 2 included a 2-hour virtual team training simulation where students embodied avatars of their health profession for real-time, virtual interprofessional rounds in two different scenarios. Debriefing after each scenario but no details provided. Scenario 1 was a bedside round of an elderly patient following surgery. Scenario 2 was a discussion with the patient's family regarding discharge. | Only the full team training intervention significantly improved mean team performance scores (p<0.05). Both intervention groups significantly improved mean interprofessional attitude scores (ATIHCT: p<0.05; ISVS: p<0.001). No differences between intervention groups on mean team performance (p = 0.96) and interprofessional attitude (ATIHCT: p = 1.00; ISVS: p = 0.77) scores. |
| Mahramus, 2016 [45] Dates not reported. United States | Pre/post test | **Setting:** medical simulation **laboratory** at a large teaching **hospital Population:** Hospital **code teams:** physicians, nurses and respiratory therapists **Gender:** Female: 70% m **Age:** Not reported | **Team tool:** Y **Program evaluation:** N | N | Two-hour training session included two video-recorded cardiac resuscitation scenarios on airway and cardiac arrythmia management with a high-fidelity mannequin and a 45-minute educational session covering teamwork behaviours related to leadership, communication, role and responsibility designation, and mutual respect for physicians, nurses, and respiratory therapists on code teams. A 10-minute debriefing session led by the trained simulation leader followed the intervention. The intervention focussed on teamwork during cardiac arrest. Debriefing after each scenario. | Mean scores for teamwork and the overall rating of teamwork increased after the didactic training between the simulation 1 and 2 (p < .001). No differences were noted between professional groups for the overall rating of teamwork. Respiratory therapists rated teamwork higher than physician residents in the second simulation for items related to global perspective and prioritizing tasks (p = .05). Respiratory therapists scored higher than nurses and physician residents on team morale and following standards and guidelines (p = .05). Participants identified that the debriefing sessions were helpful to reinforce learning and provide an opportunity to step back from fast-paced events for an overview of critical events. |

**Table 1.** (Continued)

| First Author, Year, Data Collection, Country | Study Design | Population Characteristics | Instruments Y if validated | Frame-work | Intervention | Results |
|---|---|---|---|---|---|---|
| Marshall, 2009 [39] Dates Not reported Australia | RCT | **Setting:** Not reported **Population**: final-year medical students **Gender:** Not reported **Age**: Not reported | Patient satisfaction questionnaire with attending rounds: N | N | Forty-minute small-group training session related to the Identification, Situation, Background, Assessment, Recommendation (ISBAR) communication tool and a simulated scenario using a patient simulator into a 2- to 4-hour didactic lecture for final year medical students. Students needed to ask a senior clinician for assistance over the telephone during a crisis situation. The 40-minute small group teaching session focussed on the importance of effective communication during telephone referrals, critique of videos exemplifying poor communication including the lack of explicit declaration of identity and location, presentation of the ISBAR tool and role plays. Students were allocated to groups of 10 to 12 participants and reporting was done by one student to represent the group. | The content and clarity of the telephone communication was rated higher ($p < 0.001$). |
| Monash, 2017 [46] September to November 2013 United States | Cluster-RCT | **Setting:** internal medicine teaching service **Population**: team of physician, senior resident ($2^{nd}$ or $3^{rd}$ year of residency training), 2 interns, and a $3^{rd}$ and/or $4^{th}$-year medical student. AND their patients admitted to the medicine service. **Gender:** Providers: *Not reported* Patients (Women) n (%):I: 301 (51); C: 337 (56) **Age** mean (SD): Providers: *Not reported* Patients: I : 59.5 (18.9); C : 60.1 (18.7) | Patient and provider questionnaires adapted from the literature: No details provided: N | N | Standardized bedside rounds to present and discuss patients' plans of care. Rounds were conducted by teams of physicians, senior residents, two interns and third- and fourth-year medical students and the patients they followed who were admitted to a medicine service. The training session lasted 1.5 hours and focussed on a bundled set of five key attending rounds recommendations that included 1) pre-round huddle to establish round schedule and priorities, 2) conduct of round, 3) inclusion of bedside nurses in rounds, 4) real-time order entry, and 5) updated patient care plan on a whiteboard. Monthly training sessions were conducted with physicians and physician residents in the intervention arm. Patient views were measured to determine their level of involvement in decision-making, quality of communication between the patient and the medical team, and perception that the medical team cared for them. | Significant differences were noted for pre-round huddle ($p < .001$), conduct of rounds ($p < .001$), inclusion of nurses ($p < .001$), real-time order entry ($p < .001$), use of whiteboard ($p < .001$). Patient satisfaction significantly different with rounds ($p = 0.011$) and perception that team cares for them ($p = 0.031$). The time spent per patient increased by four minutes on average ($p < .001$). Several differences were noted in trainee and MD satisfaction scores. Although the intervention decreased the time needed for rounding by an average of 8 minutes ($p = 0.52$), trainees perceived that attending rounds lasted longer ($p < .001$). |

*(Continued)*

**Table 1.** (Continued)

| First Author, Year, Data Collection, Country | Study Design | Population Characteristics | Instruments Y if validated | Frame-work | Intervention | Results |
|---|---|---|---|---|---|---|
| O'Leary, 2010 [47] August-February 2008 United States | RCT | **Setting :** nursing station of a tertiary-care teaching hospital **Population :** physician residents, nurses **Gender** Women n (%): *Physician Resident*: C: 25 (61), I: 23(49) *Nurses* : C: 22 (88), I: 31(91) **Age** Mean (SD) *Physician Resident*: C: 27(1.7) I: 27.6 (2.1) *Nurses*: C: 33.6 (8.3) I: 30.8 (8.0) | **Safety Attitudes Questionnaire:** Y Scale reliability was 0.9 using Raykov's ρ coefficient, indicating strong reliability [87]. | N | Structured daily inter-disciplinary rounds (SIDR) that included a structured communication tool to address the needs of newly admitted patients, patient safety and develop a plan of care. The tool was used in conjunction with regular interdisciplinary rounds co-led by the nurse manager and unit medical director. A work group met weekly over 12 weeks prior to implementation to determine content areas and develop the communication tool. | The intervention lasted an average 33.5 minutes (standard deviation: 5.7 minutes) and included a structured communication tool to discuss the needs of patients admitted within the last 24 hours. Differences noted in teamwork climate (p = .01) with higher perceptions of teamwork climate by nurses (p = .005), nurses' rating of the quality of communication and collaboration and perceptions of SIDR (p = .02). No differences noted in physician resident ratings of the quality of communication and collaboration and perceptions of SIDR. Physicians and nurses agreed that SIDR improved efficiency of the workday, collaboration and patient care. No differences were noted for safety climate, length of stay or costs. |
| Oner, 2018 [48] April-July 2016. United States | RCT | **Setting:** labor and delivery and postpartum units **Population**: nurses **Gender** n(%): *Female* I: 34 (100) C: 36 (100) **Age** mean (SD): I: 42.4 (12.3) C: 43.4 (11.3) | **Modified Pian-Smith grading scale:** Y | N | Three-hour simulation-based educational intervention on assertiveness and advocacy training for nurses. The intervention included a review of information about the Maternal Abnormal Vital Signs (MAViS), training in the Assertiveness/ Advocacy/CUS/ two-challenge rule (AACT) for nurses in labour and delivery and postpartum care to encourage speaking up, and debriefing using a two-on-one advocacy-inquiry non judgemental technique. Training included Power Point slides and pre-scripted role-playing scenarios, simulations and debriefing. Ten to 15 minute debriefing sessions performed immediately to demonstrate assertiveness, recognize emergency situations and reflect on performance and provide guidance to change internal dialogue and encourage nurses' willingness to speak up. Each simulation lasted 5–10 minutes followed by 10–15 minute debriefing. | No significant differences in speaking up were found between the control group and intervention. Differences were found within groups where nurses in labour and delivery spoke up more than nurses in post partum (2.29 ± 0.89 vs. 1.25 ± 0.43, P < 0.006). These differences remained significant after controlling for baseline differences. |

*(Continued)*

**Table 1.** (Continued)

| First Author, Year, Data Collection, Country | Study Design | Population Characteristics | Instruments Y if validated | Frame-work | Intervention | Results |
|---|---|---|---|---|---|---|
| Thomas, 2010 [49] AND Katakam, 2012 [50] June 2007 to June 2008 United States | RCT | **Setting:** Surgical and Clinical Skills Center. Cardiac arrest simulation theaters.m **Population**: incoming interns (1ˢᵗ year) for pediatrics with no previous completed NRP certification Gender: Not reported **Age**: Not reported | Neonatal Resuscitation Program Megacode Assessment Form: N | N | Two-hour training session for first-year incoming interns in pediatrics about teamwork, resuscitation using high-fidelity and low fidelity mannequins to standard neonatal resuscitation program (NRP) and debriefing after each scenario. The intervention provided by two trained instructors included information about human error, communication behaviours (information sharing, assertion, inquiry, vigilance, leadership), standard terminology, SBAR (Situation, background, assessment and recommendation) communication, customized video-clips and role playing to illustrate teamwork, and debriefing after each scenario. | Interns who received a brief teamwork curriculum with NRP training used more frequent teamwork behaviors (p = 0.001). Each additional assertion behaviour per minute (e.g., voicing an opinion, change of phase in resuscitation) resulted in a 41 second decrease in resuscitation duration (p = 0.009). Teams who received team training took less time to complete the resuscitation scenarios (p = 0.009) and resuscitation workload was better managed (p<0.001). The effect of the intervention on team behaviors persisted for at least six months (9 = 0.030). There was no clear affect on team vigilance as all teams maintained their vigilance for at least 95% of the scenario. High or low fidelity did not influence NRP performance of resuscitation duration. No information provided about debriefing. |
| Weaver, 2010 [51] February to July 2008 United States | Quasi-experimental mixed-model design | **Setting:** OR service line with a control location. C and I groups located at separate campuses **Population :** surgeons, certified registered nurse anesthetist, nurse, surgical technician anasthesiologist, physician assistant **Gender:** not reported **Age:** mean 36–55.5 years | Trainee reactions to training session: N Medical Performance Assessment tool: N Hospital Survey on Patient Safety Culture (**HSOPS**): Y HSOPS subscales between 0.40–0.83 [88] Operating Room Management, Attitudes Questionnaire (**ORMAQ**): Y | Y | Four-hour training session using interactive role playing for three interdisciplinary teams from the operating theater to improve teamwork and highlight impact of the TeamSTEPPS program. Didactic sessions included TeamSTEPPS competencies related to structured communication (e.g., SBAR, Call-Out, Check-Back), leadership, mutual support, and situation monitoring. Behaviours in the OR were measured using an observation tool to capture precase briefing and debriefing. | Differences were noted in communication (p < .05), precase briefing (p < .001), mutual support (p < .05), and situation monitoring (p < .01). No differences were noted in leadership, debriefing, dimensions of the Hospital Survey on Patient Safety and the Operating Room Management Attitudes questionnaire. |

**Table 1.** (Continued)

| First Author, Year, Data Collection, Country | Study Design | Population Characteristics | Instruments Y if validated | Frame-work | Intervention | Results |
|---|---|---|---|---|---|---|
| Weller, 2014 [41] Dates Not reported New Zealand | Pre/post test | **Setting :** post-anaesthesia care unit (PACU) simulated crisis of two major teaching hospitals **Population :** anaesthetists, anaesthetic technicians, and PACU nurses **Gender:** not reported **Age:** not reported | TeamSTEPPS Survey: Y HSOPS questionnaires: Y Reliability estimates Cronbach alpha from 0.88–0.96 reported elsewhere [86, 89]. | N | Video-recorded teaching simulation based on the Stop; Notify; Assessment; Plan; Priorities; Invite ideas (SNAPPI) structured communication tool, and a 10-minute debriefing session for anesthetists. Anesthesia technicians and post-anesthesia care unit nurses received relevant information probes about the surgical case that were not provided to the anesthetists. The intervention lasted 45 minutes and included a 15-minute baseline video-recorded simulation to explain SNAPPI and a demonstration of a simulated patient crisis. A 10-minute educational debriefing session highlighted crisis management principles. A follow-up simulation was completed an average of 37 days apart (range 24–91 days). | Anesthetists learned all the probes in 27% of simulations (range: 10–49%). Significant differences were noted for the SNAPPI scores ($p < .001$), verbalize diagnoses ($p = .043$). No differences were noted for team information sharing and medical management. The debriefing sessions highlighted that anesthetists believed that it was common for operating room personnel to have different information about a surgical case. |
| Zausig, 2009 [40] 2003 Germany | RCT | **Setting:** Two university hospitals and 5 community hospitals. Setting of scenarios: not reported **Population:** Anaesthesiologist (more than 6 months experience) **Gender** n male/female I: 10/10; C: 12/10 **Age** (years): I: 33 (30–37); C: 31 (29–35) | ANTS: Y | N | Intervention for anesthesiologists with at least six months work experience that lasted 3.5 hours and incorporated two scripted simulation scenarios and a single in-depth debriefing session to compare the medical management and non-technical skills in a simulated anesthesia crisis. Each group had a distinct debriefing strategy with an emphasis on reflecting on one's performance. The intervention included a single training session and a 30-minute video-based debriefing where medical management was addressed in both groups (10 minutes) and non-technical skills were addressed in the intervention group. The first scenario included actors and interactive lectures on topics related to crisis management and non-technical skills (i.e., resource management, planning, leadership, communication). | The overall quantity of the non-technical skills were different between the groups ($p = 0.02$). The medical management activities and the quality of the non-technical skills were highly correlated ($r = 0.59$, $p < .001$). However, the overall quality of the non-technical skills was not significantly different between the groups. A single debriefing session did not improve non-technical skill performance. |

## Settings

Studies were conducted in acute care in-patient settings including the operating room/post-anesthesia care unit [40–42, 44, 51], emergency department [52, 54], intensive care unit transport team [53], medical/surgical units [39, 58], labour and delivery [48], and crisis/cardiac arrest [43, 45, 49, 50, 55–57]

## Participants

A total of 6338 participants were included in the studies. Of these, 2955 were in the control group and 3049 in the intervention group. One study did not provide the number of participants in the intervention and control groups [58]. Three studies [41, 44, 45] included a pre-post design and only reported the total number of participants in the study. Researchers reported on studies with physicians only [40, 41, 52], medical residents/students/interns only [39, 42, 49, 50, 54, 56], nursing and medical students [43], nurses only [44, 48], nursing students only [55], and a mix of care providers [45, 47, 51, 53, 57, 58]. Only one study reported on a mix of care providers and patients [46].

Gender of providers was reported in 13 studies [40, 42–45, 47, 48, 52, 54–58] with more women in eight studies [42, 44, 45, 47, 48, 55, 57, 58], more men in two study [52, 54], and approximately equal representation of men and women in three studies [40, 43, 56]. Gender of patients reported as approximately half for women [46] in one study and mostly men in one study [54]. Gender was not reported for providers in 6 studies [39, 41, 46, 49–51, 53].

The age of providers was reported in ten studies [40, 42, 47, 48, 51, 52, 54–57] and ranged from 16 years to over 60 years in the identified studies. The age of providers was not reported in nine studies [39, 41, 43–46, 49, 50, 53, 58]. Patients with medical conditions were aged on average 59.5 years (standard deviation: 18.9) [46] and patients treated for trauma were aged 43–45 years [54].

## Instruments

A range of validated and non-validated instruments were reported. No instrument clearly stood out but the Anesthetists Non-Technical Skills (ANTS) was used in three studies [40, 43, 53]. The ANTS was used to measure task management, teamwork, situational awareness and decision-making [40, 43, 53]. Jankouskas [43] reported Cronbach α values ranging from 0.66 to 0.83 for the ANTS. Eight studies included instruments that were developed or adapted for the study including the assessment of technical skills [55], simulation scenario checklists [52, 56], team leadership and patient care measure [54], computer experience questionnaire [44], team performance [58], program evaluation [45], trainee reactions to training session, and the Medical Performance Assessment tool [51].

Nine distinct instruments were used to assess teamwork. Beck et al. [57] used the German version of the Team Assessment Scale where raters assessed team performance on three sub-scales (Cronbach α 0.67–0.81[59]). Coppens et al. [55] included a mix of self-reported and assessor evaluated instruments with Cronbach α values ranging from 0.76–0.90 for the Team Efficacy and the General Self-Efficacy Scales, and the Clinical Teamwork Scale (CTS) (Kappa .78; interclass correlation .98) [55]. Kalisch [44] included the Nursing Teamwork Survey to measure trust, team orientation, backup, shared mental model and team leadership [44]. Teamwork knowledge test examined using an eight-item test consistent with the study's conceptual framework [60]. Internal consistency (Cronbach α = 0.94) and test-retest reliability (Cronbach α = 0.92) were excellent [44]. No additional evidence of validity provided by Kalisch et al. (2015) [44]. Liaw et al. [58] included the Attitudes Towards Interprofessional Health Care Teams (Cronbach α: 0.82.) and Interprofessional Socialization and Valuing Scale (Cronbach α: 0.95). Mahramus [45] incorporated the TEAM Tool that includes 11 items to examine teamwork skills during resuscitation. Internal consistency ranged from .94 to .97. Weller [41] integrated the TeamSTEPPS and the Hospital Survey on Patient Safety Culture (HSOPS) questionnaires. No psychometric assessment provided in the paper.

Additional validated instruments were identified. Barzallo Salazar [42] used two validated instruments to measure how individuals make decisions (i.e., General Decision Making Scale

and Self-Construal Scale). No psychometric properties provided by the authors. O'Leary [47] included the Safety Attitudes Questionnaire (SAQ). The SAQ has demonstrated internal consistency, test-retest reliability, and convergent validity. No specific values provided in the text. Acceptable to excellent values for Cronbach α and inter-rater reliability reported by the authors. Patient and provider questionnaires were adapted from the literature to measure satisfaction with bedside rounds but no psychometric assessment was provided [46]. Oner et al. [48] used the modified Pian-Smith grading scale is a 5-point instrument to measure facial expression and body language to represent saying and doing nothing to advocating and inquiring repeatedly. Inter-rater agreement after training was 100%.

## Frameworks

Five studies were supported by a theoretical or a conceptual framework [43, 44, 51, 52, 57]. Authors identified 1) the Salas framework to highlight team leadership, orientation, performance behaviours and backup behaviours[44, 57]; 2) team training and social learning theory to provide both declarative knowledge and implementation examples and teach the knowledge, skills, and attitudes [52]; 3) team effectiveness conceptual framework to represent the behavioural, cognitive, and affective domains [43]; 4) a multi-level training evaluation framework to examine trainee reactions and learning, on the job behaviours, and results [51].

## Characteristics of the Interventions

**Dose.** *Duration*. Thirteen studies included interventions that lasted two hours or less [39, 41, 42, 44–47, 49, 50, 52, 55–58]. Six studies included interventions that lasted up to four hours [40, 43, 48, 51, 53, 54].

*Frequency*. Interventions were delivered in a single session [39–41, 43–45, 48–52, 54–58] or as part of daily rounds [46, 47]. Interventions could be spread over two days to one week [42] or over three months [53]. Three studies included an email reminder and a follow-up simulation to determine if changes in behaviour were sustained over time [41, 48, 51]. Two studies incorporated monthly refresher sessions [46, 53].

*Mode of delivery*. Interventions were delivered on-site [42, 46, 47, 53, 61] or outside of the usual place of work [40, 41, 43–45, 48–52, 54–58] when specific equipment or additional space was needed. Different formats were used including scenarios with actors [40, 42, 48, 53, 54], mannequins [39, 43, 45, 49, 50, 52, 54–58], video-recorded scenarios [41, 43, 45, 52], didactic material [39, 43, 52, 54, 58], podcast and a virtual environment [44, 58], focus group discussions [53]; structured communication tools [46, 47, 58]; and role playing [39, 48, 51].

**Type of Interventions.** *Simulations*. Simulation was the most frequently proposed intervention. Sixteen studies were identified [39–45, 48–50, 52–58]. High- and low-fidelity simulations were conducted for technical and non-technical skills to review basic surgical techniques and surgical errors [42], transport for critical care patients [53], resuscitation [40, 43, 45, 49, 50, 52, 55–57], leadership training [54], a virtual environment to resolve day-to-day conflicts in nursing teams [44], assertiveness training [48], and structured communication [41, 58]. Scenarios were delivered either all at once or broken down into several sessions (up to 3). The high-fidelity sessions required more extensive preparation ahead of the simulation.

*Communication*. Communication included structured communication and speaking-up.

Structured communication was included in seven studies [39, 41, 46, 47, 51, 54, 58]. Interventions included Situation, Background, Assessment, Recommendation (SBAR) [39, 49, 50, 54, 58] and the Stop; Notify; Assessment; Plan; Priorities; Invite ideas (SNAPPI) structured communication [41], standardized interprofessional bedside rounds to present and discuss patients' care plans [46, 58], interdisciplinary rounds co-led by the nurse manager and

**Table 2. Risk of bias assessment for included studies.**

| First Author (Year) | Cohort | Control /comparisongroup | Pre/post intervention data | Random assignment of participants to intervention | Random selection of participants for assessment | Follow-up rate of 80% or more | Comparison groups equivalent on socio-demographics | Comparison groups equivalent at baseline on disclosure | Total | Additional Comments |
|---|---|---|---|---|---|---|---|---|---|---|
| Barzallo Salazar (2014) [42] | 0 | 1 | 0 | 1 | 0 | 1 | 1 | NR | 4 | Students were blinded to the focus of the study. Assessor same surgeon in experimental and control groups |
| Beck (2019) [57] | 0 | 1 | 0 | 1 | 0 | 0 | 1 | NR | 3 | 15 teams in Intervention group and 27 teams in the Control group excluded from the analysis. Instructors knew about study goal. Rater blinded to group allocation. |
| Chang (2019) [53] | 0 | 1 | 1 | 1 | 0 | 1 | 1 | 1 | 6 | |
| Coppens (2018) [55] | 0 | 1 | 0 | 1 | 0 | 1 | NR | NR | 3 | |
| Evain (2019) [56] | 0 | 1 | 0 | 0 | 0 | 1 | NR | NR | 2 | Instructor embedded in scenario. No details provided for assessor training. Assessors blinded to group allocation. |
| Fernandez (2013) [52] | 0 | 1 | 0 | 1 | 0 | 1 | 1 | 1 | 5 | Data coders were blinded to condition assignments and study hypotheses |
| Fernandez (2020) [54] | 0 | 1 | 1 | 1 | 0 | 0 | 1 | 1 | 5 | Authors note that trauma team members have the potential be in both the control and intervention groups. Assessors blinded to group allocation. Assessor training clearly detailed. |
| Jankouskas (2011) [43] | 0 | 1 | 1 | 1 | 0 | 1 | 1 | 1 | 6 | Interrater reliability using intraclass correlation (one-way random effects model) between the two blinded raters was 0.90. |
| Kalisch (2015) [44] | 0 | 1 | 1 | 0 | 0 | 0 | 0 | NR | 2 | Sixteen participants completed both the pre- and the post-test. Data were analyzed by descriptive statistics (means, standard deviation, and percentages) and paired t test. |
| Liaw (2019) [58] | 0 | 1 | 0 | 0 | 0 | 1 | 1 | 0 | 3 | No information provided on how randomization was done. Assessor training indicated. |
| Mahramus (2016) [45] | 0 | 1 | 1 | 0 | 0 | 1 | 0 | NR | 3 | |
| Marshall (2009) [39] | 0 | 1 | 0 | 0 | 0 | 1 | 0 | NR | 2 | One of the senior investigators was involved in scenarios. Blinded assessors |
| Monash (2017) [46] | 0 | 1 | 0 | 0 | 0 | 0 | 1 | NR | 2 | No blinding of attending MDs and trainees. Auditors blinded to study arm allocation Data from one clinician who crossed over was removed |

*(Continued)*

**Table 2.** (*Continued*)

| First Author (Year) | Cohort | Control /comparison group | Pre/post intervention data | Random assignment of participants to intervention | Random selection of participants for assessment | Follow-up rate of 80% or more | Comparison groups equivalent on socio-demographics | Comparison groups equivalent at baseline on disclosure | Total | Additional Comments |
|---|---|---|---|---|---|---|---|---|---|---|
| O'Leary (2010) [47] | 0 | 1 | 0 | 0 | 0 | 1 | 1 | NR | 3 | The structured communication tool was used in SIDR for all patients newly admitted to the unit (admitted in previous 24 hours). The daily plan of care for all other patients (those who were not newly admitted to the unit) was also discussed, but without the aid of a structured communication tool. This decision was made by the working group in an effort to balance effective communication among providers with work efficiency. Medical director documented case discussions. Unclear who documented attendance for each discipline |
| Oner (2018) [48] | 0 | 1 | 1 | 1 | 0 | 0 | 1 | NR | 4 | Study about nurses but no nurse is part of the research team. Assessors were blinded |
| Thomas (2010) [49] | 0 | 1 | 0 | 1 | 0 | 0 | 0 | NR | 2 | Assessors are blinded for megacode and 6 month follow-up |
| Katakam (2012) [50] | | | | | | | Secondary analysis Original study is Thomas | Secondary analysis Original study is Thomas | | Unclear how coders were trained. Two additional research nurses served as performance observers and were also blinded to participant team training status. Their training consisted of approximately 40 hours each during the 6-month training period. |
| Weaver (2010) [51] | 0 | 1 | 0 | 0 | 0 | 0 | 0 | 1 | 2 | |
| Weller (2014) [41] | 0 | 1 | 1 | 1 | 0 | 1 | NA Pre-/posttest | 1 | 5 | Trained, blinded raters scored the SNAPPI in baseline and follow-up simulations against a pre-defined scoring rubric on an eight-point scale Two raters external to the study and blinded to the intervention, and to baseline or follow-up. |
| Zausig (2009) [40] | 0 | 1 | 1 | 1 | 0 | 1 | 1 | 1 | 6 | Assessors blinded |
| **Scores:** | 0/19 | 19/19 | 8/19 | 11/19 | 0 /19 | 12 /19 | 11 /19 | 7/19 | | |

physician incorporating a structured communication tool to address the needs of newly admitted patients [47], interactive role playing and didactic training to improve interprofessional teamwork in the operating theater [51].

Speaking up was included in two studies [42, 48]. In the Barzallo Salazar [42] study, the senior surgeon created an environment conducive to speaking up by encouraging trainees to speak up using a scripted scenario. In the Oner [48] study, nurses received assertiveness and advocacy training to determine if it influenced their speaking up behaviours.

*Leadership training.* One study [54] focussed on leadership training for physician residents in trauma care. The single, four-hour session included facilitated discussion of trauma leadership skills (30–45 min), a 30-minute didactic session describing leadership behaviors in trauma care, simulations, and debriefing. Simulations adapted to each participant's learning needs while meeting curriculum requirements [62]. During the simulation, each participant functioned as the team leader, while the second participant observed using the leadership checklist. Debriefing occurred immediately after each simulation. Self-identified areas for improvement and instructor observations informed subsequent simulations. An individualized learning plan was developed for each participant.

*Debriefing.* Debriefing was identified in thirteen studies in the current review [40, 41, 43–45, 48–52, 54–56, 58]. Most debriefing sessions lasted between five to 10 minutes for technical skills with the longest session lasting 30 minutes for non-technical skills. Debriefing sessions were completed immediately after the simulation scenarios in most cases and reflected key content areas (e.g., crisis management, conduct of resuscitation, teamwork behaviours, medical management). Coppens et al. [55] examined the contribution of debriefing following training on crisis resource management training, and found higher scores in the intervention group on teamwork (p = .011), team efficacy (p < .001) and technical skills (p = .014). No significant difference was noted for self-efficacy (p = 0.157) [55]. Trained facilitators were used in two studies [43, 45]. To facilitate learning, participants were provided with positive examples of teamwork behaviours [44, 45]. They were asked to reflect on what they had learned, their performance [48, 54, 55], what they would do differently in the future [44, 55]. Non-blaming techniques were specified in the Jankouskas study [43]. Debriefing was pre-recorded in two studies [40, 41]. Additionally, Zauzig [40] developed a distinct debriefing strategy for each simulation scenario. Only one session was conducted with no improvement in non-technical skill performance in the Zauzig [40] study. The level of detail of the debriefing sessions and their content was not always clearly described. It was not always possible to determine the influence of the debriefing session on participants' learning because of the limited information provided.

**Risk of bias of included studies.** The ratings for each of the eight items are described below, and results are summarized in Table 2. A summary score for each criterion is provided at the bottom of the Table 2.

Overall, the included studies met between two to six of the eight criteria. The studies by Chang et al., Jankouskas et al. and Zauzig were rated highest [36, 39, 49] [40, 43, 53]. No cohort or longitudinal study was identified in the retained studies. Eight studies reported pre/post intervention data [40, 41, 43–45, 48, 53, 54]. Eleven studies reported a random assignment of participants to the intervention [40–43, 48–50, 52–55, 57]. The other studies included a convenience sample [39, 44, 45, 51, 56, 57], random assignment at the unit level [46, 47, 57] or the process of randomization was not described[58]. Random selection of participants for assessment was not used in any of the retained studies. A follow-up and reporting rate for at least 80% of participants was achieved in twelve studies [39–43, 45, 47, 52, 53, 55, 56, 58]. Jankouskas et al. (2011) [43] reported one outcome with less than an 80% follow up (response time: oxygen placement). However, this was due to non performance of the task by the control and intervention groups rather than a risk of bias in the conduct of the study. We assessed that the researchers had met the criterion.

Further, baseline socio-demographic characteristics were provided in 14 studies [40, 42, 43, 45, 47, 48, 51–58]. No differences in baseline characteristics were reported in nine studies [40,

42, 43, 48, 52–54, 57, 58]. Two studies [45, 47] reported baseline differences between the groups. Three studies [51, 55, 56] provided baseline characteristics but no comparison between the groups. Four studies did not report any information [39, 44, 46, 49, 50]. The comparison of baseline socio-demographic information was not applicable in one study [41] because it was a pretest/posttest design. Finally, comparison groups were compared at baseline on disclosure in seven studies [40, 41, 43, 51–54].

Researchers outlined the steps taken to limit the risk of bias including participants blinded to the study purpose [42], data collectors blinded to the assignment of participants [39, 41, 43, 46, 48–50, 52, 54, 56–58], assessor training to ensure inter-rater reliability [41, 43, 49, 50, 54, 58]. Three study [43, 54, 58] reported reliability indices ranging from 0.90 to 1.0). In other studies, actions during the conduct of the study increase the risk of bias. Examples included the senior surgeon participating in the scenario destined for the control and the intervention groups [42], research team members involved in the simulation or data collection [39, 47, 56], and no clear indication of assessor training [49, 50, 56].

**Funding sources.**   Funding sources were reported in ten studies [40, 41, 44, 46, 47, 49, 50, 52–54, 58]. The role of the funder in the design, conduct or reporting of the research project was reported in two studies [46, 52]. Authors generally reported no conflict of interest except for [54] who reported that some co-authors had potential conflicts of interests.

## Discussion

The purpose of the systematic review was to identify the characteristics of brief interventions that were known to be effective to clarify roles of healthcare team members and improve team functioning. Our review highlights that research into brief team interventions is emerging as an important topic internationally. In our study sample, brief team interventions were developed to address issues in hospital settings with a range of providers including physicians, physician residents, nurses, nursing assistants, respiratory therapists, nursing and medical students, and patients in a medical ward and trauma care. No studies were conducted in primary care. High-fidelity simulations were conducted for technical skills in the operating theater and code teams to simulate a cardiac arrest or other types of crisis situations for patients. These studies required extensive preparation, highly specialized environments, and extensive resources. Structured communication and speaking-up were used for non-technical skills and required less preparation before study initiation but more sustained follow-up over the course of the study. Leadership training for non-technical skills as a short team intervention appears promising. Studies examining non-technical skills can be conducted in the teams' usual work environment. Single training sessions can be used to improve technical skills. However, single debriefing sessions may be insufficient to improve non-technical skills. Our findings extend the review findings of Marlow et al. (2017) [26] who examined effective team training interventions but did not identify short interventions.

Only two studies included patients and providers to examine the effectiveness of the intervention. Guler et al. (2017) [63] highlighted that patient experience is a key indicator to team performance. Our study highlights that there is a clear need for studies focusing on brief team interventions to clarify roles and improve team functioning that include patients and families as part of the healthcare team. White et al. (2018) [64] argued that most healthcare teams face important challenges because team membership changes across rotations and shiftwork. It is thus imperative for teams to focus on communication and clarifying roles of team members, including the roles of patients and families.

Several studies included in the review were at high risk of bias. This is concerning as it represents a threat to internal validity. Only three studies were at low risk of bias. Keeping this in

mind, some characteristics appear to show promise. Two- to four-hour sessions appear reasonable to engage provider participation. Training providers for technical skills using two-hour sessions followed by feedback appears to improve skill level, task management and performance in situations such as cardiac arrests or crisis situations in the operating theater. Training for non-technical skills including communication, care coordination, understanding one's role and the role of others in the team (role clarity) appears to require more time with 4-hour training. Three to four training sessions lasting 30 minutes to one hour spread out over several weeks with structured facilitation and debriefing appear to improve the use of non-technical skills. Monthly meetings appear to sustain change over time. A recently published feasibility study by Fontenot and White (2019) [65] examining moral distress of nurses in the intensive care unit included an intervention with four 30-minute debriefing sessions every two weeks. The authors assessed that the intervention was feasible and acceptable in a busy work environment, and the debriefing improved non-technical skills related to self-awareness and management of moral distress. The cost and availability of replacement personnel for trainees are additional factors to consider when planning training sessions.

Simulation-based training followed by debriefing sessions provides a safe setting for healthcare professionals to develop non-technical skills. Debriefing is a key element when using simulation-based studies to enhance learning and self-awareness [66]. However, one debriefing session does not improve performance of non-technical skills. Previously, didactic methods of training and video-based learning were mostly used to hone technical and non-technical skills of healthcare providers away from clinical environment [67]. Gradually, as the need to mimic the clinical setting increases, simulation settings must evolve rapidly to provide a more realistic experience for learners and include patients in simulations and debriefings. It is particularly important to plan debriefing sessions using a debriefing framework [68] and consider including patient actors in the debriefing sessions. Low-fidelity simulation may be more beneficial when limited resources are available. In addition, in-situ training is necessary to investigate feasibility of implementing team skills in a clinical environment where the challenges of the healthcare system reside [67]. Although different simulation training methods have been utilized to demonstrate the significance of acquiring teamwork competencies among healthcare members, there remains a gap in translating the outcomes of simulation training in the clinical setting.

It is imperative to transfer the outcomes of team interventions from simulation settings to clinical environments [69]. Despite efforts to demonstrate the effect of simulation on improving non-technical skills, it continues to be a challenge [69]. In the studies mentioned in this systematic review, various brief team interventions were implemented in different settings and measured using diverse validated and non-validated instruments. Thus, it is crucial to develop brief team interventions based on theoretical constructs of team functioning measured using conceptually coherent validated instruments that appropriately evaluate different aspects of brief team interventions in a simulation-based and in-situ settings.

Some studies were excluded from the systematic review even though the intervention lasted less than four hours (e.g., [70–73]). As highlighted by Fiscella et al. (2016) [74] teams in sports and in primary care share several challenges (e.g., role clarity, communication) to improve team performance, yet the most prominent among them is to align teamwork competencies and clinical practice requirements of providers. An important consideration in the decision to retain an article in our systematic review was the ability to translate the interventions to the healthcare context. Seidl (2017) [72] attempted to develop team skills using LEGO serious play in an academic setting [72]. Prichard et al. (2007) [75] proposed to work on team skills by building an AM radio. Dalenberg et al. (2009) [70] examined the contribution of military cadets discussions of a team strategy to identify and disable an adversary. Volpe et al. (1996)

[73] examined how training and workload while flying a fighter jet in a simulator influenced team processes. Cannon-Bowers et al. (1998) [71] built on the Volpe [73] study to understand how cross-training for young Navy recruits who needed to monitor a radar screen improved their ability to distinguish quickly between hostile and non-hostile contacts. These findings were difficult to apply to healthcare teams but they may provide different strategies to consider to improve team functioning and team performance.

## Limitations

Some limitations need to be kept in mind with the current review. We searched extensively for published and unpublished RCTs with no restrictions on language or geography. However, we may have missed studies because of the lack of standardized terminology in this emerging area. The quality of reporting was an important consideration in our review. In many cases, researchers did not adequately describe the study participants (e.g., age, profession, gender) or the intervention. Using reporting guidelines (e.g., CONSORT 2010) will promote the completeness and accuracy of study reporting [76]. More complete reporting of participants' gender would allow for the determination of intervention effects according to gender.

As indicated above, several studies were at an increased risk of bias. Although we reviewed additional literature to assess the instruments used in the included studies, incomplete reporting made it difficult to accurately assess some studies for risk of bias. Similarly, we may have scored debriefing sessions at an increased risk of bias due to incomplete descriptions of the sessions and the use of training debriefing instructors. More rigorous studies are needed using validated tools to measure outcomes as well as the inclusion of a theoretical or conceptual framework to guide study conduct. Careful consideration needs to be given to when to use of high-fidelity simulations given the prohibitive costs of the material and resource intensive preparation to conduct high quality simulations. Our results indicate that low-fidelity simulations may be an appropriate intervention for the acquisition on non-technical skills.

## Future research

Our review identified three key knowledge gaps where additional research is needed. Subsequent research needs to examine the effectiveness of interventions in teams in primary care, the inclusion of patients and families and evaluating short team interventions in different settings. We identified one study using simulation training for nurses working in a correctional facility [77]. The study was excluded from our review because it did not meet all of our eligibility criteria. Subsequent research needs to focus on areas outside the hospital setting. Interventions in primary care teams are needed because these teams are structured differently than teams in acute care and they may have different priorities. Fleury et al. (2019) [78] completed a cross-sectional survey of mental health teams (n = 315) in primary and specialized care, and found that team attributes (e.g., type of professional, recovery promotion) had a greater impact on team functioning in primary care teams while team processes were more important in specialized care teams. As argued by Marriage et al. (2016) [2] current team assessment tools are based on judgments of observable behaviours because they provide a quantifiable account of team performance. Future research also needs to focus on measuring the processes of teamwork rather than solely the outcomes of teamwork [2, 79, 80]. The inclusion of patients and families at all stages of the intervention's development and the evaluation of the intervention's impact is essential in the context of patient centered care. Finally, the inclusion of arts or serious play methodology in the development of brief interventions may support the emergence of creative solutions to enhance team functioning.

## Conclusion

We conducted a systematic review to determine the characteristics of brief interventions to clarify roles and improve functioning in healthcare teams. We identified 19 experimental and quasi-experimental studies that tested interventions lasting less than half a day or five hours. High-fidelity simulations were used to develop technical skills to manage cardiac arrests and crisis situations. These sessions were shorter but required more extensive preparation. Structured communications required longer sessions with participants but may be more effective to develop non-technical skills. Debriefing can be used to support the acquisition of technical and non-technical skills. Incomplete reporting of study information was found in several studies and risk of bias was assessed as high for several studies in our sample. Intervention characteristics that appear to influence successful outcomes include using three to four 30 to 60 minutes sessions spread over two to four weeks and debriefing with a trained facilitator. Monthly follow-ups appear to sustain change over time for non-technical skills. Additional research is needed in primary care and with patients and families. We anticipate that these brief interventions can be implemented on a large scale in healthcare teams to support role clarification for patients, families and providers.

## Supporting information

**S1 Data. PRISMA 2009 checklist.**
(DOC)

**S1 File. Pubmed search strategy.**
(DOCX)

## Author Contributions

**Conceptualization:** Kelley Kilpatrick, Lysane Paquette, Eric Tchouaket, Nicolas Fernandez, Marie-Dominique Beaulieu, Carl-Ardy Dubois.

**Data curation:** Kelley Kilpatrick, Lysane Paquette, Mira Jabbour, Grace Al Hakim, Véronique Landry, Nathalie Gauthier, Marie-Dominique Beaulieu, Carl-Ardy Dubois.

**Formal analysis:** Kelley Kilpatrick, Lysane Paquette, Mira Jabbour, Eric Tchouaket, Nicolas Fernandez, Véronique Landry, Marie-Dominique Beaulieu, Carl-Ardy Dubois.

**Funding acquisition:** Kelley Kilpatrick, Mira Jabbour, Eric Tchouaket, Nicolas Fernandez, Véronique Landry, Nathalie Gauthier, Marie-Dominique Beaulieu, Carl-Ardy Dubois.

**Investigation:** Kelley Kilpatrick, Lysane Paquette, Mira Jabbour, Carl-Ardy Dubois.

**Methodology:** Kelley Kilpatrick, Lysane Paquette, Mira Jabbour, Eric Tchouaket, Marie-Dominique Beaulieu.

**Project administration:** Kelley Kilpatrick, Mira Jabbour.

**Software:** Mira Jabbour.

**Supervision:** Kelley Kilpatrick.

**Validation:** Kelley Kilpatrick, Grace Al Hakim, Nathalie Gauthier, Marie-Dominique Beaulieu, Carl-Ardy Dubois.

**Writing – original draft:** Kelley Kilpatrick, Lysane Paquette, Mira Jabbour, Eric Tchouaket, Nicolas Fernandez, Grace Al Hakim, Nathalie Gauthier, Marie-Dominique Beaulieu, Carl-Ardy Dubois.

**Writing – review & editing:** Kelley Kilpatrick, Lysane Paquette, Mira Jabbour, Eric Tchoua-ket, Nicolas Fernandez, Grace Al Hakim, Véronique Landry, Nathalie Gauthier, Marie-Dominique Beaulieu, Carl-Ardy Dubois.

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
