## [Decision Letter · Decision Letter 0]

21 Apr 2020

PONE-D-20-06599

Systematic review of the characteristics of brief team interventions to clarify roles and improve functioning in healthcare teams

PLOS ONE

Dear Dr Kilpatrick,

Thank you for submitting your manuscript to PLOS ONE. After careful consideration, we feel that it has merit but does not fully meet PLOS ONE’s publication criteria as it currently stands. Therefore, we invite you to submit a revised version of the manuscript that addresses the points raised during the review process.

Wile both reviewers saw merit in the manuscript, a minor number of issues were raised.

We would appreciate receiving your revised manuscript by Jun 05 2020 11:59PM. To enhance the reproducibility of your results, we recommend that if applicable you deposit your laboratory protocols in protocols.io, where a protocol can be assigned its own identifier (DOI) such that it can be cited independently in the future. For instructions see: http://journals.plos.org/plosone/s/submission-guidelines#loc-laboratory-protocols

We look forward to receiving your revised manuscript.

Kind regards,

César Leal-Costa, Ph. D

Academic Editor

PLOS ONE

2. We noted that your article search was carried out more than a year ago. Please update your search to include recently published articles.

Reviewers' comments:

Reviewer's Responses to Questions

**Comments to the Author**

1. Is the manuscript technically sound, and do the data support the conclusions?

Reviewer #2: Yes

Reviewer #3: Yes

2. Has the statistical analysis been performed appropriately and rigorously? 

Reviewer #2: N/A

Reviewer #3: Yes

3. Have the authors made all data underlying the findings in their manuscript fully available?

Reviewer #2: Yes

Reviewer #3: Yes

4. Is the manuscript presented in an intelligible fashion and written in standard English?

Reviewer #2: Yes

Reviewer #3: Yes

5. Review Comments to the Author

Reviewer #2: Authors hypothesize that improving healthcare workers' understanding of their roles within teams will result in better healthcare delivery performance and patient safety. Authors conduct a systematic literature search/review (1990-2019) of previously-published work conducted (by other researchers) related to short-duration (less than half of one day) medical team training activities. Analysis, description, and discussion focus on 15 papers reporting on 14 different studies from five different countries. These studies investigated a variety of different short-term training modalities. The total combined number of healthcare workers across all 14 studies was 5730 (2761 assigned to control groups, and 2842 assigned to treatment groups). Authors here use the combined positive findings from these 15 papers to infer/highlight the potential for short-duration team training approaches to improve healthcare team performance.

I believe the contribution of this manuscript is in facilitating an introduction on the topic of short-duration team training for medical teams for those who are not familiar with the existing literature. Readers who are interested in improving healthcare team performance can save time in studying the literature on short-duration approaches by reading this survey first. Personally, I also thought that the discussion of benefits and trade-offs of including patient simulation was interesting, and a more in-depth discussion of integrating this approach with alternatives could be a worthwhile topic for future work.

Going forward, I suggest that the work of Herbert Clark on "Joint Activity Theory," is useful/relevant, and could potentially be considered for the creation of a scientific foundation for this general, and currently fragmented, topic. [see, Herbert Clark (1996), "Using Language," ISBN-13: 978-0521567459, ISBN-10: 0521567459, https://www.amazon.com/Using-Language-Herbert-H-Clark/dp/0521567459 ]

Reviewer #3: After article valuation “Systematic review of the characteristics of brief team interventions to clarify roles and improve functioning in healthcare teams”, I think that the current manuscript structure is very good.

The introduction is comprehensive and justifies the study in an appropriate way. However, I miss information detail related to team characteristics for effective teamwork. I recommend include some lines regarding this issue.

I recommend reviewing the exclusion criteria in order to clearly explain the reason for excluding studies with qualitative methodology because this methodology can be used to measure satisfaction and mental model.

Finally, line 176 indicates “training sessions were conducted with all assessors (n=8) to review inclusión and exclusion criteria, the screening instrument and answer questions”. The role of the eight evaluators in the research is not clear to me because it is indicated that the review is performed by two reviewers. Additionally, the authors worked with an academic librarian to establish an effective search strategy for each database. Please clarify this point.

6. PLOS authors have the option to publish the peer review history of their article (what does this mean?). If published, this will include your full peer review and any attached files.

Reviewer #2: No

Reviewer #3: No

---

## [Author Response · Author response to Decision Letter 0]

21 May 2020

Dr Leal-Costa and Reviewers, 

Thank you for your insightful comments. All the changes have been made in the manuscript entitled Systematic review of the characteristics of brief team interventions to clarify roles and improve functioning in healthcare teams (PONE-D-20-06599) as requested. The changes are summarized in the table below. 

Reviewers' comments: Response Page(s)

We noted that your article search was carried out more than a year ago. Please update your search to include recently published articles. Database searches updated to April 21st 2020. Five additional papers identified and added to the review. Changes made throughout the text

Comments to the Author 

 Comments Reviewer #2: 

Going forward, I suggest that the work of Herbert Clark on "Joint Activity Theory," is useful/relevant, and could potentially be considered for the creation of a scientific foundation for this general, and currently fragmented, topic. [see, Herbert Clark (1996), "Using Language," ISBN-13: 978-0521567459, ISBN-10: 0521567459, https://www.amazon.com/Using-Language-Herbert-H-Clark/dp/0521567459 ] Thank you for the recommendation, we will look into adding Herbert Clark’s work as proposed. 

Comments from Reviewer #3: 

I miss information detail related to team characteristics for effective teamwork. I recommend include some lines regarding this issue. Description of effective teams added Line 41-46

I recommend reviewing the exclusion criteria in order to clearly explain the reason for excluding studies with qualitative methodology because this methodology can be used to measure satisfaction and mental model. Sentence changed to:

The primary aim of the review was to identify effective short team interventions. As proposed by Higgins et al. (2019), we excluded observational and longitudinal studies and as well as qualitative methodologies as these studies are at increased risk of bias [32]. Line 125-128 

line 176 indicates “training sessions were conducted with all assessors (n=8) to review inclusión and exclusion criteria, the screening instrument and answer questions”. The role of the eight evaluators in the research is not clear to me because it is indicated that the review is performed by two reviewers. We corrected our mistake. Thank you for noting. 

Training sessions were conducted with all assessors (n=8) to review inclusion and exclusion criteria, the screening instrument and answer questions Line 140-141

Additionally, the authors worked with an academic librarian to establish an effective search strategy for each database. Please clarify this point. Text changed to: 

… academic librarian to develop and validate the search strategy and identify keywords for each database. Line 105-106

---

## [Decision Letter · Decision Letter 1]

27 May 2020

Systematic review of the characteristics of brief team interventions to clarify roles and improve functioning in healthcare teams

PONE-D-20-06599R1

Dear Dr. Kilpatrick,

We are pleased to inform you that your manuscript has been judged scientifically suitable for publication and will be formally accepted for publication once it complies with all outstanding technical requirements.

With kind regards,

César Leal-Costa, Ph. D

Academic Editor

PLOS ONE

Additional Editor Comments (optional):

Reviewers' comments:

Reviewer's Responses to Questions

**Comments to the Author**

1. If the authors have adequately addressed your comments raised in a previous round of review and you feel that this manuscript is now acceptable for publication, you may indicate that here to bypass the “Comments to the Author” section, enter your conflict of interest statement in the “Confidential to Editor” section, and submit your "Accept" recommendation.

Reviewer #3: All comments have been addressed

2. Is the manuscript technically sound, and do the data support the conclusions?

Reviewer #3: Yes

3. Has the statistical analysis been performed appropriately and rigorously? 

Reviewer #3: Yes

4. Have the authors made all data underlying the findings in their manuscript fully available?

Reviewer #3: Yes

5. Is the manuscript presented in an intelligible fashion and written in standard English?

Reviewer #3: Yes

6. Review Comments to the Author

Reviewer #3: (No Response)

7. PLOS authors have the option to publish the peer review history of their article (what does this mean?). If published, this will include your full peer review and any attached files.

Reviewer #3: No

---

## [Editor Report · Acceptance letter]

29 May 2020

PONE-D-20-06599R1 

 Systematic review of the characteristics of brief team interventions to clarify roles and improve functioning in healthcare teams 

Dear Dr. Kilpatrick:

I am pleased to inform you that your manuscript has been deemed suitable for publication in PLOS ONE. Congratulations! Your manuscript is now with our production department. 

With kind regards,

on behalf of

Dr. César Leal-Costa 

Academic Editor

PLOS ONE